

# Effects of arbuscular mycorrhizal fungi and earthworms on plant-soil systems of urban turf grasses

Ying Hou, Zongying Zhang, Yue Yuan, Xinyu Zhu and Jinping Ding

Department of Surveying and Planning, Shangqiu Normal University, Shangqiu City, Henan Province, China

## ABSTRACT

**Background**. Arbuscular mycorrhizal fungi (AMF) and earthworms have been known to enhance plant growth and improve soil quality, but the results have shown some inconsistencies, exhibiting synergistic or independent effects under different studies.

**Methods**. In this study, we conducted a factorial experiment to investigate the effects of arbuscular mycorrhizal fungi and earthworms, both individually and in combination, on plant-soil systems of urban turf grasses. Five turf grass species of different functional groups (gramineous and leguminous) were inoculated with AMF (*Glomus mosseae*) and earthworms (*Pheretima tschiliensis* Michaelsen, 1928) in a pot culture experiment, and we comprehensively assessed the eco-physiological properties of both the plant and the soil.

**Results**. Inoculation with AMF and earthworms promoted the growth of five turf grass species. Specifically, AMF had a greater impact on the height and biomass of leguminous plants (white clover and alfalfa) compared to gramineous plants (ryegrass, early meadow grass, and tall fescue), while the effects of earthworms on two functional groups were opposite to those of AMF. AMF and earthworms had different dimensions of effects on the physical and chemical properties of the soil compared to the control, with AMF showing a better improvement in soil quality than earthworms. The results indicated that effects of AMF and earthworms on urban turf grasses and soil were dependent on different species. Dual inoculation with AMF and earthworms showed positive effects exclusively on the soil properties of five turf grass species, but only positive on the growth of gramineous plants. The comprehensive evaluation indicated that dual inoculation with AMF and earthworms did not improve the plant-soil system more effectively than AMF alone. Conversely, gramineous plants inoculated with AMF showed the greatest effective improvement in the soil-plant system. Therefore, dual inoculation with AMF and earthworms did not have a more positive effect on plant-soil system than that could be expected from the effect of single inoculation of either AMF or earthworms. Further analysis showed that AMF exhibited the most comprehensive improvements in the plant-soil system of gramineous plants, indicating that the effects of AMF and earthworms on the soil-plant system of lawn might be influenced by the specific plant species.

Corresponding author
Jinping Ding, jinpingding@163.com

## INTRODUCTION

Urban lawns can offer important ecological services, such as carbon sequestration, heat island reduction, and increased water infiltration (*Wheeler et al., 2017*). However, intensive urban lawn maintenance compared to trees and shrubs such as mowing, irrigation, fertilization and use of pesticides may lead to a range of issues such as disrupted soil structure, severe compaction, nutrient imbalance, elevated soil pH levels and pollution (*Gómez-Brandón et al., 2022*; *Xia et al., 2023*). There exists a reciprocal relationship between soil and plant (*Wang, Wang & Liu, 2021*), highlighting the impact of urban soil on plant growth in urban green spaces, thus influencing the overall landscape aesthetics and ecological benefits of urban greening efforts. The above issues will affect the growth of lawn grass and even the ecological functions of the urban lawn. Therefore, it is fundamental to find ways to improve urban soil quality and promote lawn grass growth for the ecological function of urban lawns.

Various methods can be employed to improve urban soil, including soil replacement, application of organic and chemical fertilizers, soil conditioners, beneficial bacteria, *etc*. While, chemical fertilizers are commonly used, their application can lead to soil pollution and ecological harm (*Zhang & Xiao, 2022*). In recent years, the use of biotechnology as a sustainable improvement technique to improve soil has received greater attention (*Elhindi, El-Din & Elgorban, 2017*). Numerous studies have explored the application of Arbuscular mycorrhizal fungi (AMF) and earthworms on plants, focusing on their effects on growth, resistance, yields, and soil properties (*Enkhtuya, Rydlova & Vosatka, 2000*; *Zhang et al., 2017*; *Liu et al., 2020*). Arbuscular mycorrhizal fungi are beneficial microorganisms widely existing in soil. The symbiotic relationships between plants and AMF have received increasing attention worldwide because of its significant contribution to plant growth, nutrient acquisition, soil carbon sequestration and plant tolerance to stress (*Thirkell, Pastok & Field, 2020*; *Chandrasekaran & Paramasivan, 2022*; *Hawkins et al., 2023*; *Li et al., 2025*). Particularly on poor soil or in the stress environment, the application of AMF can have a significant positive effect on the uptake of soil nutrients and enhancing stress tolerance (*Wagg et al., 2014*; *Meng et al., 2021*). Arbuscular mycorrhizal fungi improve rhizosphere micro-environment, establish a rich mycelial network around the plant root, and the huge mycelium can act as an extension of the root system, capturing more nutrients and water for the plant, thus improving plant growth and enhancing the host plant resistance to abiotic and biotic stresses (*Meng et al., 2021*; *Xie et al., 2021*; *Cheng et al., 2021*; *Li et al., 2024*). Earthworms live in soil, feeding on live and dead organic matter and playing a crucial role as decomposers in ecosystems (*Zhang et al., 2016*; *Wang et al., 2020*). Earthworms can modify the structure, microbial activity and nutrient mineralization in soils (*Brussaard et al., 2007*; *Muchane et al., 2018*; *Mahohi & Raiesi, 2021*), contributing to soil fertility through activities such as feeding, digesting, excreting, and burrowing (*Hu et al., 2021*; *Lu et al., 2022*). However, the contribution of earthworms to soil structure varies with their ecological strategy. Endogeic earthworms live in the upper layer of the mineral soil and feed on soil enriched with organic matter. They make horizontal burrows and are considered major agents of aggregation and soil organic matter stabilisation, compared with
epigeic earthworms that live in the organic layer at the soil surface and rarely make burrows (*Muchane et al., 2018*). Earthworms can also have an effect on the plant growth through changing the spatiotemporal availability of carbon (C), nitrogen (N) and phosphorus (P) nutrients in their casts and burrow walls (*Muchane et al., 2018*; *Li et al., 2019*). However, investigations about the role of AMF and earthworms have predominantly concentrated on crops, overlooking urban lawn, which can provide multiple ecosystem services to humans and the environment. Few studies have validated the role of AMF and earthworms on lawn, especially on different urban turf grasses.

Urban lawns are dominated by one or a few perennial grass species, usually belonging to Poaceae, Cyperaceae or Leguminosae. They are often considered as simplified ecosystems with low plant diversity and require intensive maintenance (*McKinney, 2008*; *Wheeler et al., 2017*). These managements with mowing, watering, fertilizing and use of pesticides may damage the quality of the lawn soil. Soil structure was often spoiled, arrangement of horizons is disturbed, chemical properties vary greatly, and soil biodiversity and fertility drop. But their management and remediation are often overlooked in urban environments (*Wang et al., 2003*). Therefore, the effect of AMF and earthworms on urban lawn and soil remain poorly understood. On the one hand, most of the previous studies just evaluated effects on plants or soil in isolation, without considering the holistic plant-soil systems. On the other hand, while both AMF and earthworms individually are known to benefit soil or plant growth, their combined effects may differ from their individual effects on plants and soils. Previous studies on the interactions between AMF and earthworms have shown varying results, with synergistic or independent effects observed (*Tuffen, Eason & Scullion, 2002*; *Wurst et al., 2004*; *Eisenhauer et al., 2009*). This inconsistency suggests that the effects of earthworms and AMF on plants and soil may be influenced by soil characteristics or the plant species under study (*Wurst et al., 2004*). Therefore, more research is still needed to enrich the understanding of the role of AMF and earthworms on plant-soil systems.

We aimed at investigation the effect of AMF and earthworms on different urban turf grasses growth and lawn soil properties. Previous studies indicated that AMF and earthworms not only promoted the growth of plants but also improved soil quality (*Zhang et al., 2016*; *Li et al., 2024*). We expected that inoculation of AMF and earthworms would significantly improve the holistic plant-soil system of lawn significantly. However, emerging evidences demonstrate differential effects on plant growth-promoting or improvements in soil properties of AMF and earthworm across plant species (*Wurst et al., 2004*; *Cui et al., 2020*; *Chen et al., 2023*). As such, we hypothesized that the responsiveness of urban turf grasses to inoculation with AMF and earthworms exhibited species-specific variation. We further hypothesized that dual inoculation of AMF and earthworms could have a more positive effect on plant-soil system than that could be expected from the effect of single inoculation of either AMF or earthworms. To test these hypotheses, five commonly used turf grasses, including two leguminous plants: alfalfa (*Medicago sativ* a L.) and white clover (*Trifolium repens* L.) and three gramineous plants: ryegrass (*Lolium perenne* L.), early meadow grass (*Poa pratensis* L.) and tall fescue (*Festuca arundinacea* Schreb.), were selected as research subjects. They were individually inoculated with AMF, earthworms, and a combination of both to assess the effects on plant growth and soil properties.

A comprehensive evaluation equation was constructed using principal component analysis and the weighted subordinate function method to evaluate the effects of AMF and earthworms on the plant-soil system.

## MATERIALS & METHODS

### Experimental materials

The arbuscular mycorrhizal fungi strain used was *Glomus mosseae* (BEG167), provided by Shi Zhaoyong's group at Henan University of Science and Technology, and AMF inoculums consisted of spores (about 28 g$^{-1}$), hyphae and colonized root fragments. Earthworms (endogeic: *Pheretima tschiliensis* Michaelsen, 1928) used in the experiment were collected from the same location two weeks before the experiment, which lived in the upper layer of the mineral soil and feed on soil enriched with organic matter. They make horizontal burrows and are considered major agents of aggregation and soil organic matter, thus modulating organic matter decomposition by ingesting it and excreting it in a more stable form *Zhu et al. (2025)*. Before adding, earthworms, with similar fresh weights and lengths, were washed with sterilized water, placed in sterilized plastic Petri dishes for 48 h, and then washed again to remove mycorrhizal spores as possible from their body surface and intestinal tract. The tested turf grasses selected for the study included three gramineous plants (*Festuca arundinacea* Schreb., *Poa pratensis* L., Lolium *perenne* L.) and two leguminous plants (*Medicago sativa* L., *Trifolium repens* L.). The soil characterized by a pH of 7.78, an organic matter mass fraction of 7.67 g kg$^{-1}$, an available phosphorus mass fraction of 15.84 mg kg$^{-1}$, and an available potassium mass fraction of 70.46 mg kg$^{-1}$. The potting test was carried out in April-May 2022 at the experimental base of Shangqiu Normal University. Each planting pot, with an upper caliber of 17 cm and a height of 15 cm, was filled with two kg of sterilized soil.

### Experimental methodologies

The experiment consisted of four treatments: (1) control (CK, no AMF inoculation, no earthworms), (2) AMF (inoculation with AMF), (3) EW (inoculation with earthworms, and (4) AMF+EW (dual inoculation with both AMF and earthworms), with fifteen replicates for each treatment, totaling 60 pots. 10g of AMF inoculum were placed at five cm soil depth, and mixed with grass seeds at sowing. Thus half of the pots ($N = 30$) were added AMF inoculums. Meanwhile, 10 g of autoclaved (121 °C; 1 h) AMF inoculum was added to the control and EW treatments to ensure the basic consistency of the soil microbial communities. Based on the previous studies and our previous experiment (*Boag et al., 1994*; *Zhu et al., 2016*), we determined the number of individual earthworms to be added (five earthworms) in each pot of the earthworm-inoculated treatment, and the top of the pots were sealed with gauze to prevent the earthworms escaping. According to the density of the artificial lawn, 30 grass seeds were sown in each pot. After seedling emergence, 20 seedlings per pot were planted, and sampling was conducted after 30 d of plant growth.

## Measurements
### Detection of the levels of mycorrhizal colonization
Thirty root fragments of each plant were randomly selected, cut to one cm in length, stained, prepared and examined microscopically using the staining method of *Philips & Hayman (1970)*. Root mycorrhizal colonization degree was assessed as the percentage of root lengths colonize by mycorrhizal fungi against total observed root lengths.

$$\text{Mycorrhizal infection rate} = \frac{\text{The length of root infected with AMF}}{\text{Total length of root checked}} * 100\%.$$

### Growth parameters
The plants were carefully removed from the pots, and then the roots were washed to remove soil particles under slow-running water. The plants were separated into roots and shoots. Plant height and root length were measured respectively. Roots and shoots were dried at 100 °C for at least 72 h and weighed, then the root to shoot ratio was calculated according to the method described by *Zhang et al. (2015)*.

$$\text{Root/shoot} = \frac{\text{root weight}}{\text{shoot weight}}.$$

### Analysis of soil
Soil bulk density was measured by cutting ring method, soil field capacity by indoor experimental method and mass water content of soil by oven drying method. Soil porosity was calculated from soil bulk density and specific gravity. Soil pH was measured in deionized water by pH meter (Mettler Toledo FE28), electrical conductivity by a conductivity meter (Mettler Toledo FE38), and soil organic matter content was measured using the potassium dichromate oxidation method with external heat. The methods used were carried out following the methods reported by *Bao (2013)*.

### Comprehensive evaluation
Principal component analysis (PCA) was performed on 11 indicators, including four indicators of plant growth and seven indicators of soil properties, and the principal components were extracted according to the eigenvalue and cumulative contribution rate. The formula of principal component calculation was constructed, and the scores of the principal components of each process were calculated using the fuzzy subordinate function method. The formula for the fuzzy subordinate function calculation was as follows:

$$U(X_i) = \frac{(X_i - X_{min})}{(X_{max} - X_{min})} \quad (i = 1, 2, 3, 4...n) \tag{1}$$

$$U(X_i)_R = 1 - \frac{(X_i - X_{min})}{(X_{max} - X_{min})} \quad (i = 1, 2, 3, 4...n) \tag{2}$$

where: $U(X_i)$ is the value of the subordinative function of indicator $i$; $X_i$ is the measured value of indicator $i$; $X_{max}$ and $X_{min}$ are the minimum and maximum values of indicator

*i*, respectively. In this study, plant height, root length, biomass, soil porosity, soil field capacity, mass water content, and organic matter content were calculated using Eq. (1), and root-to-shoot ratio, soil bulk density, soil pH, and electrical conductivity were calculated using Eq. (2).

The expression of comprehensive evaluation was constructed by weighted summation, and a comprehensive score was calculated to characterize the combined effects of the treatments on plants and soil.

### Data analysis

Effects of AMF and earthworms on growth (plant height, root length, biomass, root-to-shoot ratio) of urban turf grasses and soil (soil porosity, soil field capacity, mass water content, soil bulk density, soil pH, electrical conductivity and organic matter content) were analyzed using two-way analysis of variance (ANOVA) in SPSS 23.0. And all data were tested for normality and homogeneity of variance (Levene's test, $P > 0.1$) before data analysis. Data would be converted if normality and homogeneity of variance were not met. Multiple comparisons using the Least Significant Difference (LSD) method determined differences in each variable among the four treatments. The principle component analysis (PCA) was performed in SPSS 23.0 to give the characteristic root and variance contribution of the principal component factor. And calculations of formulae analyses were performed in Microsoft Excel 2019.

## RESULTS

### Effects of AMF and earthworms on the growth of five turf grass species

#### Plant height and root length of five turf grass species

Compared with the control, the inoculation with AMF, earthworms and the dual inoculation significantly increased the plant height of five turf grass species (Fig. 1), but the increase varied with treatments and plant types. AMF increased the plant height of leguminous plants by 42.68% ($P < 0.05$, $F = 9.465$) and gramineous plants by 32.25% ($P < 0.05$, $F = 10.542$), and the increase in plant height of legume was significantly greater than that of gramineae. Earthworms increased the plant height of legumes by 27.94% ($P < 0.05$, $F = 6.764$), and gramineous plants by 45.08% ($P < 0.05$, $F = 5.427$) compared to the control treatment. The dual inoculation with AMF and earthworms resulted in substantial increased in plant height for ryegrass, early meadow grass and tall fescue by 60.21% ($P = 0.05$, $F = 6.562$), 47.52% ($P < 0.001$, $F = 35.116$) and 57.73% ($P < 0.001$, $F = 38.638$) respectively, which were higher than that of inoculation with AMF or earthworms alone. The plant heights of white clover and alfalfa increased by 42.17% and 43.15% by dual inoculation with AMF and earthworms, respectively. The results showed that AMF promoted legume height more than gramineae, while earthworms showed the opposite trend. Meanwhile, there was a significant interactive effect between AMF and earthworms in plant height of gramineae, but not in legumes. Furthermore, the dual inoculation with AMF and earthworms positively impacted the height of gramineae plants but not of legumes.

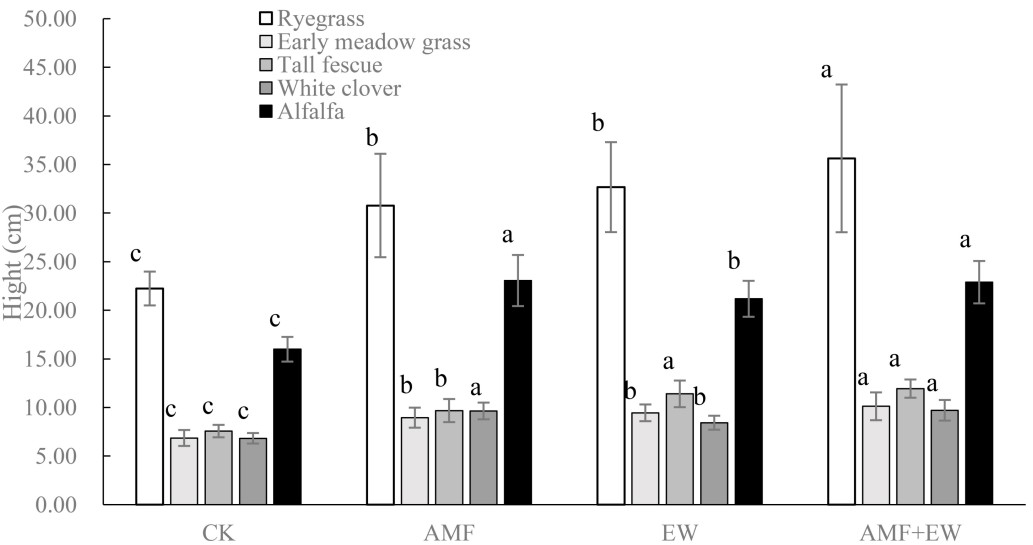

**Figure 1** **Effects of AMF and earthworms on plant height(cm) of five turf grass species.** CK: control treatment; AMF: inoculation with AMF; EW: inoculation with earthworms; AMF+EW: dual inoculation with AMF+earthworms. Different letters above each column indicate a significant difference among treatments ($P < 0.05$). The vertical bars represent the standard error of the means.

Root length of turf grasses was significantly higher for AMF, EW and AMF+EW than for CK. There was a significantly interactive effect of AMF+EW on root length. AMF significantly promoted the root length of turf grasses except for alfalfa, with increases of 43.57% ($P < 0.001$, $F = 37.524$), 37.69% ($P = 0.002$, $F = 15.517$), 36.00% ($P < 0.001$, $F = 30.185$) and 21.00% ($P = 0.032$, $F = 7.284$) observed in ryegrass, early meadow grass, tall fescue and white clover, respectively. These findings suggested that AMF had a greater role in promoting the root lengths of gramineae plants (Fig. 2). Earthworms increased the root length of five turf grass specie by 17.44% ($P = 0.005$, $F = 15.237$), 23.01% ($P = 0.011$, $F = 10.246$), 26.24% ($P < 0.001$, $F = 32.571$), 41.70% ($P = 0.002$, $F = 21.354$) and 33.38% ($P < 0.001$, $F = 38.256$) in ryegrass, early meadow grass, tall fescue, white clover and alfalfa, respectively. This indicated that earthworms had a greater promoting effect on the root length of the two leguminous plants compared to the three gramineous species. The dual inoculation treatment had a positive interaction in promoting root length of five turf grass species. The root length increments of ryegrass, early meadow grass, tall fescue, white clover and alfalfa were 58.09% ($P < 0.001$, $F = 38.415$), 47.71% ($P < 0.001$, $F = 32.718$), 39.17% ($P = 0.026$, $F = 6.245$), 51.64% ($P < 0.001$, $F = 34.723$) and 38.89% ($P = 0.007$, $F = 8.942$), respectively, which were greater than that of the inoculation with AMF or earthworms alone.

### Biomass and its allocation of five turf grass species

Treatments AMF, EW and AMF+EW significantly increased the biomass of gramineae and legume species compared to CK (Fig. 3). An interaction was observed between AMF and earthworms on the biomass of gramineae species, but not on legume. The biomass of ryegrass, early meadow grass, tall fescue, white clover and alfalfa significantly increased

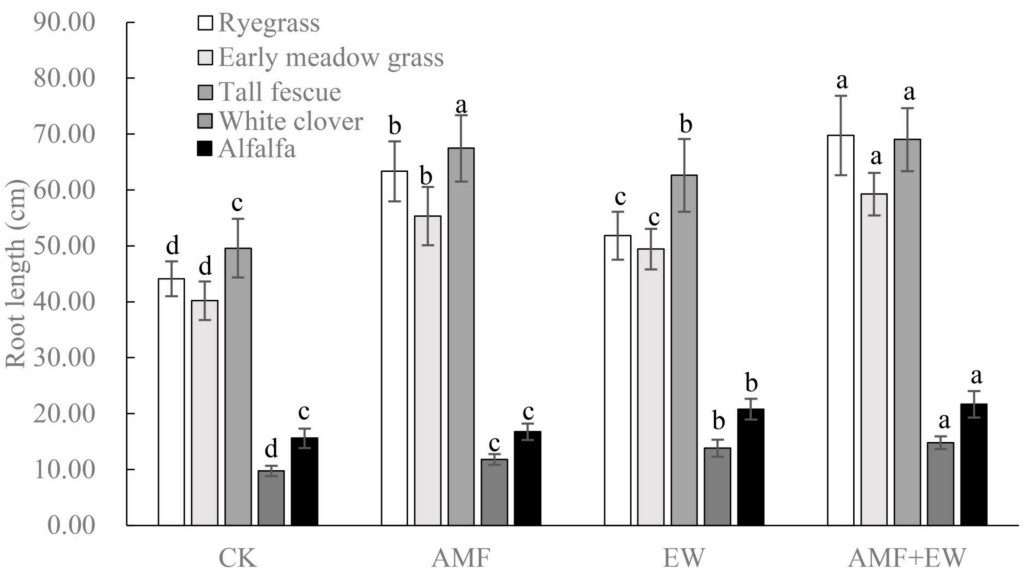

**Figure 2** **Effects of AMF and earthworms on root length (cm) of five turf grass species.** CK: control treatment; AMF: inoculation with AMF; EW: inoculation with earthworms; AMF+EW: dual inoculation with AMF+earthworms. Different letters above each column indicate a significant difference among treatments ($P < 0.05$). The vertical bars represent the standard error of the means.

by 29.05%, ($P = 0.015$, $F = 8.756$), 32.65% ($P < 0.001$, $F = 30.721$), 24.07% ($P = 0.006$, $F = 11.154$), 40.63% ($P < 0.001$, $F = 34.083$) and 46.11% ($P < 0.001$, $F = 39.425$) under AMF treatment respectively. These results suggested that AMF had a greater impact on enhancing the biomass of legume compared to gramineae species (Fig. 3). Conversely, earthworms inoculation had the opposite effect to AMF, with ryegrass, early meadow grass and tall fescue experiencing biomass increases of 41.91%, 44.90% and 45.06% ($P < 0.001$), while white clover and alfalfa showed increases of 22.73% ($P = 0.002$, $F = 13.474$) and 15.54% ($P = 0.024$, $F = 6.347$), suggesting that the earthworms promoted the biomass of gramineae plants more than that of legume. The biomass of ryegrass, early meadow grass and tall fescue increased by 62.66% ($P = 0.049$, $F = 5.046$), 59.18% ($P < 0.001$, $F = 30.475$) and 61.11% ($P < 0.001$, $F = 35.739$) in the dual inoculation treatment, which was higher than that of inoculation with AMF or earthworms alone. However, the dual inoculation did not have a significant interaction on the two leguminous plants, with the biomass of white clover and alfalfa biomass increased by 46.02% ($P = 0.075$, $F = 3.241$) and 49.22% ($P = 0.282$, $F = 1.421$), respectively. Further analysis showed that the dual inoculation with AMF and earthworms had a positive interactive effect on the biomass of gramineae plants.

Root-to-shoot ratios of gramineae plants were significantly higher for treatments AMF and AMF+EW, but lower for EW, than for CK (Table 1). There was a significant ($P < 0.05$) interactive effect of AMF and earthworms on root-to-shoot ratios of gramineae plants. AMF promoted ryegrass, early meadow grass and tall fescue to allocate more biomass below ground, leading to significant increases in root-to-shoot ratios of 17.89% ($P = 0.036$,

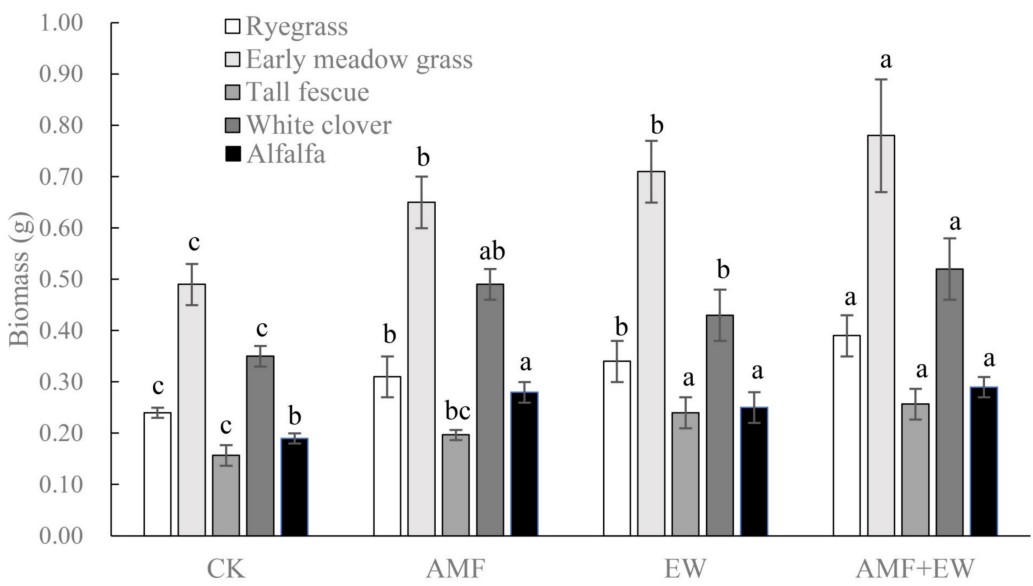

**Figure 3** **Effects of AMF and earthworm s on plant biomass (g) of five turf grass species.** CK: control treatment; AMF: inoculation with AMF; EW: inoculation with earthworms; AMF+EW: dual inoculation with AMF+earthworms. Different letters above each column indicate a significant difference among treatments ($P < 0.05$). The vertical bars represent the standard error of the means.

$F = 6.725$), 27.73% ($P < 0.001$, $F = 37.534$) and 13.92% ($P = 0.008$, $F = 20.427$), whereas alfalfa allocated more biomass to the above ground, and thus the root-to-shoot ratios decreased by 17.79% (Table 1). Contrary to AMF, earthworms promoted the ryegrass, early meadow grass and tall fescue to distribute more biomass to the above ground, causing significant decreases in root-to-shoot ratio of 14.25% ($P = 0.013$, $F = 9.246$), 18.12% ($P = 0.005$, $F = 15.462$) and 7.04% ($P = 0.054$, $F = 4.218$). Although there were no significant effects of adding AMF or earthworms alone on the root-to-shoot ratios of leguminous plants, AMF+ EW significantly ($P < 0.05$) decreased the root-to-shoot ratios of leguminous plants. The dual inoculation of AMF and earthworms showed a significant interactive effect on the root-to-shoot ratios of leguminous plants. The results suggested that dual inoculation with AMF and earthworms facilitated the development of root systems in gramineae plants, while restrained the development of root systems in leguminous plants.

## Soil properties of five turf grass species

Inoculation with AMF or earthworms had no effects on soil organic matter content and pH of five turf grass species, but there was a significant ($P < 0.05$) interactive effect of AMF and earthworms on pH of five turf grass species (Table 2). AMF increased soil field capacity, mass water content and soil porosity of five turf grass species averagely by 9.54%, 16.37% and 8.92% significantly, and the bulk density and electrical conductivity decreased averagely by 14.10% and 17.13% ($P < 0.01$) significantly (Table 2). Earthworms increased only soil field capacity and mass water content of ryegrass, early meadow grass and tall

**Table 1   Effects of AMF and earthworms on root/shoot on five turf grass species.**

| Treatment | Ryegrass | Early meadow grass | Tall fescue | White clover | Alfalfa |
|---|---|---|---|---|---|
| CK | 35.10 ± 4.21b | 45.82 ± 4.56c | 58.21 ± 4.15b | 41.63 ± 5.62a | 56.22 ± 6.05a |
| AMF | 41.41 ± 3.65a | 58.53 ± 4.72a | 66.31 ± 7.53a | 38.34b±4.81a | 46.23 ± 3.50b |
| EW | 30.12 ± 2.84c | 37.55 ± 3.17d | 54.10 ± 6.72b | 42.16 ± 5.46a | 54.71 ± 5.82a |
| AMF+EW | 42.54 ± 3.49a | 51.91 ± 4.58b | 68.30 ± 5.47a | 36.04 ± 3.19b | 43.81 ± 5.34c |
| Significance | | | | | |
| AMF | 0.036 | <0.001 | 0.008 | 0.067 | 0.022 |
| EW | 0.013 | 0.005 | 0.054 | 0.100 | 0.120 |
| AMF+EW | <0.001 | 0.015 | <0.001 | 0.021 | <0.001 |

Notes.

The data in the table are the mean ± standard deviation, and different lowercase letters in the same column indicate significant differences ($P < 0.05$). CK: control treatment; AMF: inoculation with AMF; EW: inoculation with earthworms; AMF+EW: dual inoculation with AMF+earthworms.

fescue average by 11.12% and 15.44%, whereas white clover and alfalfa decreased averagely by 7.18% and 6.85% ($P < 0.05$) in soil bulk density and electrical conductivity. The results indicated that inoculation with AMF had a greater effect on soil improvement than with earthworms. Dual inoculation increased soil field capacity, mass water content and soil porosity averagely by 12.61%, 17.28% and 11.65% ($P < 0.05$) of five turf grass species, and soil density and electrical conductivity decreased by an average of 16.11% and 23.32% ($P < 0.001$). Overall, the magnitude of the change in dual inoculation treatment was greater than in the inoculation with AMF or earthworms alone, which showed that dual inoculation had a positive interaction on the soil properties of the five-plant species.

## Comprehensive evaluation of the effects of AMF and earthworms on plant-soil system

The PCA related to plant growth and soil properties was carried out. The first four principal components were extracted. Characteristic values were 4.82, 2.37, 1.38 and 1.10, and the variance contribution rates were 43.81%, 20.68%, 12.54% and 10.00%, respectively, thus the cumulative contribution rate was 87.03%, which indicated that the four principal components covered most of the information of soil physical and chemical properties and plant growth indicators.

Based on the characteristic roots and factor loading coefficients in Table 3, the expressions for calculating the principal component factor scores were derived from formulae. And according to these formulae, scores of the principal component factors were calculated.

$F1 = 0.18^*ZX1+0.20^*ZX2+0.16^*ZX3-0.03^*ZX4-0.40^*ZX5+0.39^*ZX6+0.32^*ZX7+0.32^*ZX8-0.28^*ZX9-0.41^*ZX10+0.37^*ZX11$

$F2 = -0.03^*ZX1+0.50^*ZX2-0.20^*ZX3+0.50^*ZX4+0.17^*ZX5-0.16^*ZX6+0.38^*ZX7+0.41^*ZX8+0.22^*ZX9+0.19^*ZX10-0.09^*ZX11$

$F3 = 0.74^*ZX1+0.07^*ZX2-0.56^*ZX3-0.28^*ZX4+0.03^*ZX5+0.01^*ZX6-0.02^*ZX7-0.05^*ZX8+0.11^*ZX9+0.16^*ZX10+0.16^*ZX11$

$F4 = 0.11^*ZX1+0.32^*ZX2+0.55^*ZX3-0.45^*ZX4+0.10^*ZX5-0.35^*ZX6+0.25^*ZX7-0.14^*ZX8+0.36^*ZX9+0.02^*ZX10+0.17^*ZX11$

**Table 2  Effects of AMF and earthworms on soil physical and chemical properties of turf grass species.**

| Species | Treatments | Soil bulk density (g cm⁻³) | Total porosity (%) | Soil field capacity (%) | Mass water content (%) | Soil pH | Electrical conductivity (μS cm⁻¹) | Soil organic matter content (%) |
|---|---|---|---|---|---|---|---|---|
| Ryegrass | CK | 1.39 ± 0.07a | 44.10 ± 5.03b | 21.50 ± 2.64b | 16.10 ± 1.02b | 7.76 ± 0.45a | 245.36 ± 20.42a | 0.79 ± 0.01b |
| | AMF | 1.18 ± 0.09b | 49.33 ± 3.07a | 22.73 ± 2.55ab | 18.42 ± 1.04a | 7.68 ± 1.01a | 203.20 ± 16.22b | 0.85 ± 0.02a |
| | EW | 1.31 ± 0.07a | 45.32 ± 3.06b | 24.21 ± 3.26a | 19.30 ± 2.04a | 7.53 ± 0.31a | 237.23 ± 18.36a | 0.81 ± 0.02b |
| | AMF+EW | 1.12 ± 0.13b | 50.67 ± 3.28a | 24.13 ± 2.01a | 18.91 ± 2.41a | 7.49 ± 0.45a | 184.63 ± 9.54c | 0.89 ± 0.03a |
| | Significance | | | | | | | |
| | AMF | <0.001 | 0.027 | 0.052 | 0.005 | 0.052 | <0.001 | 0.023 |
| | EW | 0.061 | 0.120 | <0.001 | <0.001 | 0.104 | 0.063 | 0.071 |
| | AMF+EW | <0.001 | <0.001 | 0.006 | 0.003 | 0.055 | <0.001 | 0.046 |
| Early meadow grass | CK | 1.40 ± 0.09a | 43.56 ± 5.58b | 20.84 ± 1.68b | 16.27 ± 2.18b | 7.75 ± 0.62a | 238.37 ± 19.37a | 0.80 ± 0.02a |
| | AMF | 1.11 ± 0.12b | 47.67 ± 7.21a | 23.13 ± 1.85a | 18.62 ± 1.52a | 7.71 ± 0.83a | 193.20 ± 18.57b | 0.81 ± 0.02a |
| | EW | 1.37 ± 0.08a | 44.33 ± 4.35b | 24.19 ± 1.63a | 18.25 ± 0.94a | 7.65 ± 0.57a | 227.63 ± 11.46a | 0.82 ± 0.02a |
| | AMF+EW | 1.14 ± 0.08b | 49.29 ± 5.68a | 23.53 ± 3.14a | 19.43 ± 1.72a | 7.46 ± 0.63a | 173.37 ± 9.27c | 0.83 ± 0.01a |
| | Significance | | | | | | | |
| | AMF | 0.002 | <0.001 | 0.014 | 0.021 | 0.157 | <0.001 | 0.070 |
| | EW | 0.056 | 0.067 | 0.004 | 0.035 | 0.063 | 0.138 | 0.052 |
| | AMF+EW | 0.003 | <0.001 | <0.001 | <0.001 | <0.001 | <0.001 | 0.105 |
| Tall fescue | CK | 1.36a ± 0.14a | 45.83 ± 4.16b | 20.16 ± 1.48b | 16.84 ± 2.35b | 7.69 ± 0.62a | 251.54 ± 21.72a | 0.80 ± 0.03a |
| | AMF | 1.21 ± 0.07b | 48.96 ± 5.49a | 22.27 ± 3.14ab | 19.67 ± 1.59b | 7.65 ± 0.38a | 210.13 ± 20.57b | 0.80 ± 0.01a |
| | EW | 1.32 ± 0.09a | 46.40 ± 4.59b | 24.43 ± 2.16a | 21.50 ± 1.81a | 7.72 ± 0.47a | 246.23 ± 19.36a | 0.81 ± 0.02a |
| | AMF+EW | 1.20 ± 0.11b | 50.19 ± 5.35a | 24.57 ± 2.83a | 20.73 ± 2.05a | 7.41 ± 0.41a | 195.40 ± 20.42b | 0.83 ± 0.02a |
| | Significance | | | | | | | |
| | AMF | 0.034 | 0.040 | <0.001 | <0.001 | 0.072 | <0.001 | 0.051 |
| | EW | 0.146 | 0.073 | 0.002 | <0.001 | 0.057 | 0.056 | 0.062 |
| | AMF+EW | <0.001 | <0.001 | <0.001 | 0.025 | 0.026 | <0.001 | 0.075 |
| White clover | CK | 1.39 ± 0.08a | 44.87 ± 5.71b | 19.84 ± 2.16b | 15.22 ± 1.17b | 7.58 ± 1.05a | 250.55 ± 15.45a | 0.79 ± 0.02a |
| | AMF | 1.22 ± 0.09b | 49.21 ± 3.26a | 22.16 ± 2.45a | 18.50 ± 1.15a | 7.55 ± 0.41a | 206.47 ± 12.37b | 0.81 ± 0.03a |
| | EW | 1.25 ± 0.07b | 46.52 ± 3.72ab | 20.24 ± 1.03b | 16.42 ± 1.61b | 7.61 ± 0.33a | 223.51 ± 20.45b | 0.82 ± 0.02a |
| | AMF+EW | 1.21 ± 0.07b | 50.73 ± 4.52a | 21.35 ± 1.53a | 17.23 ± 1.43a | 7.41 ± 0.28a | 198.42 ± 17.45b | 0.83 ± 0.03a |
| | Significance | | | | | | | |
| | AMF | 0.007 | <0.001 | 0.027 | <0.001 | 0.114 | <0.001 | 0.053 |
| | EW | 0.024 | 0.061 | 0.153 | 0.056 | 0.071 | 0.005 | 0.068 |
| | AMF+EW | <0.001 | <0.001 | 0.003 | 0.011 | 0.042 | <0.001 | 0.055 |
| Alfalfa | CK | 1.40 ± 0.11a | 45.26 ± 3.67b | 20.17 ± 1.45b | 16.35 ± 2.14b | 7.62 ± 0.64a | 248.75 ± 16.93a | 0.78 ± 0.01a |
| | AMF | 1.24 ± 0.05b | 48.35 ± 3.15a | 22.25 ± 2.64a | 18.74 ± 1.32a | 7.59 ± 0.57a | 210.28 ± 12.72b | 0.81 ± 0.02a |
| | EW | 1.19 ± 0.07b | 47.54 ± 4.15a | 20.92 ± 1.04b | 17.92 ± 1.56ab | 7.65 ± 0.35a | 215.13 ± 19.84b | 0.80 ± 0.01a |
| | AMF+EW | 1.15 ± 0.06b | 48.72 ± 5.23a | 21.87 ± 2.18b | 18.51 ± 2.12a | 7.39 ± 0.43a | 195.40 ± 18.27b | 0.83 ± 0.02a |
| | Significance | | | | | | | |
| | AMF | <0.001 | 0.003 | <0.001 | <0.001 | 0.101 | <0.001 | 0.071 |
| | EW | <0.001 | 0.008 | 0.065 | 0.053 | 0.064 | 0.008 | 0.101 |
| | AMF+EW | <0.001 | <0.001 | 0.054 | 0.027 | <0.001 | <0.001 | 0.063 |

**Notes.**

The data in the table are the mean ± standard deviation, and different lowercase letters in the same column indicate significant differences ($P < 0.05$). CK: control treatment; AMF: inoculation with AMF; EW: inoculation with earthworms; AMF+EW: dual inoculation with AMF+earthworms.

**Table 3  Results of principal component analysis.**

| Plant and soil indicators | Factor loading | | | |
|---|---|---|---|---|
| | First principal component (math.) | Second principal component | Third principal component | Fourth principal component |
| Plant height (X1) | 0.400 | −0.050 | 0.866 | 0.119 |
| Root length (X2) | 0.434 | 0.754 | 0.078 | 0.339 |
| Dry weight (X3) | 0.351 | −0.300 | −0.655 | 0.572 |
| Root to shoot ratio (X4) | −0.065 | 0.751 | −0.324 | −0.475 |
| Soil bulk density (X5) | −0.876 | 0.249 | 0.032 | 0.102 |
| Soil Porosity (X6) | 0.853 | −0.235 | 0.003 | −0.371 |
| Field capacity (X7) | 0.709 | 0.579 | −0.018 | 0.262 |
| Mass water content (X8) | 0.699 | 0.618 | −0.055 | −0.152 |
| pH (X9) | −0.614 | 0.338 | 0.128 | 0.380 |
| Electrical conductivity (X10) | −0.905 | 0.282 | 0.183 | 0.017 |
| Organic matter content (X11) | 0.814 | −0.140 | 0.184 | 0.182 |
| Characteristic root | 4.82 | 2.27 | 1.38 | 1.10 |
| Variance contribution | 43.81% | 20.68% | 12.54% | 10.00% |
| Weighting factor | 0.50 | 0.24 | 0.14 | 0.12 |

**Table 4  Comprehensive evaluation scores of plant growth and soil indicators under different treatments.**

| Treatment | Principal component 1 score | Principal component 2 score | Principal component 3 score | Principal component 4 score | Aggregate score |
|---|---|---|---|---|---|
| CK | 0.20 | 0.57 | −0.04 | 0.06 | 0.24 |
| AMF | 0.47 | 1.15 | 0.10 | 0.41 | 0.57 |
| EW | 0.17 | 1.05 | 0.11 | 0.18 | 0.37 |
| AMF+EW | 0.21 | 1.31 | 0.29 | 0.55 | 0.52 |

where *ZXi* denoted the data after bastardization of the initial indicator, *i.e.*, the fuzzy affiliation function value)

Then, weighted calculations based on the principal component contribution rate resulted in a comprehensive evaluation equation:

$F = 0.50{*}F1 + 0.24{*}F2 + 0.14{*}F3 + F4{*}0.12$.

The integrated evaluation scores of different treatments were shown in Table 4. The integrated scores were ranked as AMF > AMF+EW > EW > CK, indicating that the inoculation of AMF had the most significant integrated improvement on the plant-soil system, followed by the dual inoculation. Further analyses also showed that the combined improvement effect on the plant-soil system was also related to plant species. In this study, the best comprehensive improvement effect on the plant-soil system occurred under treatments of ryegrass inoculated with AMF and earthworms, followed by tall fescue and early meadow grass, while the two legumes had a relatively poor combined improvement effect.

## DISCUSSION

### Effects of AMF and earthworms on different urban turf Grasses species

The results of this study showed that both AMF and earthworms positively influenced the growth of five turf grass species, showing significant improvements in plant height, root length and biomass. This outcome was consistent with our initial expectation that the inoculation of AMF and earthworms would promote the growth of turf grasses. By supplying the necessary phosphorus for nitrogen fixation in plant roots, AMF enhanced the nitrogen fixation capacity of plants, thereby stimulating their growth (*Zhao, Li & Zhao, 2006*). Earthworms facilitated plant growth through mechanisms such as increasing nutrient availability, enhancing the population and activity of beneficial soil microorganisms, reducing pathogen populations, producing growth-promoting hormones, and modifying soil structure (*Hodson et al., 2023*). However, the effects of AMF and earthworms on the growth of different plant species varied. AMF had a more significant promoting effect on white clover and alfalfa, possibly due to the synergistic interactions between AMF and rhizobacteria. Earthworms promoted the growth of ryegrass, early meadow grass and tall fescue more significantly due to their well-developed root systems, which had higher nutrient uptake and utilization efficiency. This was consistent with the results of an earlier study (*Groenigen van et al., 2014*). The combined treatment of AMF and earthworms had a positive impact on the growth of the three gramineae species. However, this positive interaction was not observed in the two species of legume. This discrepancy can be attributed to the fact that legume is dicotyledonous plants, which is more dependent on mycorrhizae in nutrient-poor soil conditions. The activities of earthworms may have decreased mycorrhizal colonization in legumes, thereby inhibiting the mutual benefit between AMF and legumes. The above differences indicated that effects of AMF and earthworms on plants were dependent on plants species, supporting our first hypothesis.

The biomass allocation of five turf grass species changed significantly after inoculation with AMF and earthworms (Fig. 3). AMF inoculation and dual inoculation increased biomass allocation to root of three gramineous plants and led to higher root-to-shoot ratios. The present results were consistent with the study of *Wang et al. (2007)* on ryegrass, but different from the results of *Henkes et al. (2018)* on wheat. This inconsistency is likely due to different soil fertility between farmland and grassland. In infertile grassland soil, AMF presence stimulated the excretion of endogenous hormones, which could promote root development, and resulting in a higher root-to-shoot ratio (*Zhang et al., 2017*). The larger root system of the plants presumably assisted plant nutrients and water uptake. Conversely, the root-to-shoot ratio of the two legumes decreased under inoculation with AMF and dual inoculation treatments because AMF promoted nutrient uptake of the legumes, leading to the biomass being preferentially allocated to the above ground, resulting in a decrease in root-to-shoot ratio. This was consistent with the results of *Peng, He & Zhang (2021)* and *Wu et al. (2022)*. The root-to-shoot ratio of the three gramineous plants also decreased under inoculation with earthworms. It is likely because the activities of earthworms improved the soil structure and soil fertility (*Hodson et al., 2023*), and

consequently, the limitation of resources to the plant's underground organ was alleviated, as a result, the biomass was preferentially allocated to the above-ground parts, resulting in a lower root to shoot ratio.

## Effects of AMF and earthworms on soil

Consistent with our expectation, AMF and earthworms improved the soil qualities (Table 2), but AMF had better soil improvement effects than earthworms. The plant inoculated with AMF produced more endogenous hormone-indoleacetic acid (Chen et al., 2023), which stimulated the formation of more secondary roots (Table 1). Although previous studies indicated that higher root inputs increased soil organic matter content (Wang et al., 2025), we did not observe the similar results (Table 2), which might be due to the short experimental duration in this study. On the other hand, larger root systems improved the looseness of the soil, increasing the soil porosity (Table 2), thus decreasing the soil bulk density. Meanwhile, AMF treatment significantly increased soil field capacity and mass water content. The glomalin-related soil protein secreted by AMF could promote the formation of soil aggregates with stable water content, which improved soil permeability and water-holding capacity was the main reasons (Alizadeh et al., 2020). Dual inoculation with AMF and earthworms has a positive interaction on the soil of the five turf grass species. The reasons might be that the soil used in this study was the urban green-field soil, which was poor in soil nutrients, especially soil organic matter. This coincided with results from study by Li et al. (2019), reporting that the effects of AMF and earthworms might be stronger in the nutrient-poor soils where the earthworms could exert a greater effect on the mycorrhizal colonization in soils because although earthworms might graze some hyphal and even disturb the hyphal network, the addition of the earthworms resulted in a significantly higher mycorrhization percentage, either due to perturbation or improved spore germination (Pelosi et al., 2024). On the other hand, inoculation with AMF provided more food (mycorrhizal mycelia and the AMF-colonized roots) for earthworms in poor soil. Earthworms stimulated the process of soil aggregation to improve soil porosity and structure through ingesting, burrowing and faeces (Cui et al., 2020; Meng et al., 2021), and therefore, more favorable for AMF and earthworms to promote each other's effects on soils.

## Interactions of AMF and earthworms on plant-soil systems

The plant-soil system is the basic structural unit of the biosphere, and soil and plant interact and constrain each other in the ecosystem (Wang, Wang & Liu, 2021). In the present study, dual inoculation with AMF and earthworms did not show positive interactions for plant-soil system improvement compared to the effects of single inoculation with either AMF or earthworms, and thus our hypothesis was rejected. Similar results were shown by Muchane et al. (2018), who reported no positive effects of earthworms and AMF interactions on crop performance and soil nutrition. However, our result was contrast with other studies that showed positive effects of other endogeic earthworms and AMF interactions on crop growth and soil quality (Ma, Dickinson & Wong, 2006; Li et al., 2013). In the present study, although dual inoculation with AMF and earthworms has a positive interaction on the soil

of the five turf grass species, it did not exhibit a consistent positive effect on plant growth, thus affecting its comprehensive effect on the plant-soil system. Therefore, the effects on the plant-soil system of inoculation with AMF (comprehensive evaluation scores = 0.57) were more pronounced than that dual inoculation with AMF and earthworms (comprehensive evaluation scores = 0.52; Table 4). As far as we are aware, there are still no consistent results on the interaction between AMF and earthworms (*Tuffen, Eason & Scullion, 2002*; *Wurst et al., 2004*; *Milleret, Bayon & Gobat, 2009*). Presumably, the effects of interaction between earthworms and AMF on plant-soil system depend on soil characteristics, experimental organisms and the density of earthworms. We attributed the lack of such interactive effect due to soil characteristics. Our experiment was carried out in urban green soil, which was poor in soil organic matter and nutrition. Inoculation with AMF and earthworms improved the physical properties of the soil, but did not significantly alter soil chemical properties (soil pH and organic matter content; Table 2), which might affect soil nutrient availability, and thus hindered a more positive interaction of AMF and earthworms.

## CONCLUSIONS

In conclusion, AMF and earthworms promoted the growth of five turf grass species and improved the soil properties, however, the impacts varied among the different plant species, which supported the first hypothesis regarding effects of inoculation with AMF and earthworms on urban turf grasses is dependent on different species. The comprehensive assessment revealed that the combined inoculation of AMF and earthworms did not result in greater improvements in the plant-soil system compared to AMF inoculation alone, despite the observed positive interaction between AMF and earthworms in enhancing plant growth and soil properties, and thus, our second hypothesis regarding dual inoculation of AMF and earthworms could have a more positive effect on plant-soil system than that could be expected from the effect of single inoculation of either AMF or earthworms was rejected. We attributed the lack of such interactive effect due to soil poor in nutrients and organic matter. The present study also provided directly empirical evidence that AMF exhibited the most comprehensive improvements in the plant-soil system of gramineous plants.

### Funding

This study was supported by the Natural Science Foundation of Henan Province (242300420229), the Program for Science and Technology Innovation Talents in Universities of Henan Province (21HASTIT015), and the Science and Technology Project of Henan Province (172102410054). The funders had no role in study design, data collection and analysis, decision to publish, or preparation of the manuscript.

### Grant Disclosures

The following grant information was disclosed by the authors:

Natural Science Foundation of Henan Province: 242300420229.
Program for Science and Technology Innovation Talents in Universities of Henan Province: 21HASTIT015.
Science and Technology Project of Henan Province: 172102410054.

## Competing Interests

The authors have no relevant financial interests to disclose.

## Author Contributions

- Ying Hou conceived and designed the experiments, authored or reviewed drafts of the article, and approved the final draft.
- Zongying Zhang analyzed the data, prepared figures and/or tables, and approved the final draft.
- Yue Yuan performed the experiments, prepared figures and/or tables, and approved the final draft.
- Xinyu Zhu conceived and designed the experiments, prepared figures and/or tables, and approved the final draft.
- Jinping Ding performed the experiments, authored or reviewed drafts of the article, and approved the final draft.

## Data Availability

The raw data is available in the Supplemental File.

## Supplemental Information

Supplemental information for this article can be found online at http://dx.doi.org/10.7717/peerj.20289#supplemental-information.

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
