# Peer review of "Effects of arbuscular mycorrhizal fungi and earthworms on plant-soil systems of urban turf grasses"

_PeerJ, doi:10.7717/peerj.20289_

## Round 0.1 · original submission · Major Revisions

· Academic Editor

Major Revisions

Please have a look at the reviewers comments. They highlight some serious areas for improvement of your manuscript.

Reviewer 1 ·

Basic reporting

Using biological approaches to enhance soil functions is an effective method for achieving soil health. The authors demonstrated the benefits of biological interventions on soil health and plant growth by examining the effects of AMF and EW (earthworm) alone or in combination on five plant-related soil system functions. Overall, this study is worthy of publication. However, the following scientific issues need to be addressed by the authors prior to submission.

Experimental design

Clarify how AMF strains were added to the soil. Describe the methodology for the control group.

Validity of the findings

We suggest performing a two-way ANOVA, as the authors explicitly mentioned comparing interaction effects. However, the independent effects of AMF and EW, as well as their interaction effects, are not presented in the data analysis.

Additional comments

Introduction section: The beneficial effects of AMF inoculation on soil health are not sufficiently highlighted. Additionally, the authors should prioritize citing the most recent studies to reflect the timeliness of their research. We recommend referencing the following recent papers:
https://doi.org/10.1016/j.scitotenv.2024.173975
https://doi.org/10.1016/j.still.2024.106443
Both studies provide detailed discussions on AMF-mediated improvements in soil carbon and nutrient cycling, which could strengthen the authors' arguments.

Table 5: The significant change in bulk density (BD) without a corresponding shift in SOC (soil organic carbon) is surprising. The authors should address this discrepancy in the Discussion section. Reference Wang et al. (https://doi.org/10.1016/j.agee.2024.109456), which thoroughly examines correlations between soil properties, for deeper insights.
Table 3: Remove redundant parentheses. Avoid repetitive descriptions for similar tables.
Align "CK\T1\T2\T3" centrally to match the centered data below.
References: Italicize journal names consistently (some are currently italicized while others are not).

Reviewer 2 ·

Basic reporting

This article is devoted to the study of the influence of inoculation with arbuscular mycorrhizal fungi and the presence of worms on the growth and development of turf grasses. Despite the comprehensive yet not novel approach used, I have quite an ambivalent impression of this study. The Results and Discussion sections are mostly clear, while the other sections of the text contain numerous problems, which highly decrease the quality of the current manuscript. These are related mostly to the relatively poor quality of the text, unclear hypotheses, and the partly misleading Introduction. I explain my concerns in more detail throughout the Review.

General Major comments and Introduction issues:

I’ve detected minor yet consistent problems with English writing, which should be proofread throughout the Manuscript to improve its quality.

Additionally, the consistency of terms used should be verified. For example, “Arbuscular mycorrhizal fungi” can be used without a capital letter, while the AM/AMF abbreviation could be introduced and used afterwards throughout the MS. Also, I recommend that the Authors avoid simultaneously using both the abbreviation and the full name in the same sentences or throughout the text.
Introduction

LL. 73-85: This paragraph appears unexpected at the beginning of the MS, as no topic related to urbanization or urban soils was introduced in the Abstract or the title. I suggest joining this one with the third paragraph of the Introduction (LL. 100-112), and also adjusting the title + Abstract to fit the topic. I would also recommend introducing grasses (as commonly found in urban soils) and the soil used.

LL. 86-99: I can get the general idea of this paragraph well, yet I suggest adjusting and restructuring that to improve the quality. Please introduce AMF and worms in more detail. Please tell the readers how the application of AMF is beneficial, and what it means (e.g., inoculation of plants). Please provide references for every earthworm effect separately. Additionally, the justification is poorly given, and the only reference is provided. Please, have a look at the following papers:

Mahohi, A., & Raiesi, F. (2020). The performance of mycorrhizae, rhizobacteria, and earthworms to improve Bermuda grass (Cynodon dactylon) growth and Pb uptake in a Pb-contaminated soil. Environmental Science and Pollution Research, 28, 3019-3034. https://doi.org/10.1007/s11356-020-10636-z.
Muchane, M., Pulleman, M., Vanlauwe, B., Jefwa, J., & Kuyper, T. (2019). Impact of arbuscular mycorrhizal fungi and earthworms on soil aggregate stability, glomalin, and performance of pigeonpea, Cajanus cajan. Soil Research. https://doi.org/10.1071/SR18096.
Li, Y., Wang, S., Lu, M., Zhang, Z., Chen, M., Li, S., & Cao, R. (2019). Rhizosphere interactions between earthworms and arbuscular mycorrhizal fungi increase nutrient availability and plant growth in the desertification soils. Soil and Tillage Research. https://doi.org/10.1016/J.STILL.2018.10.016.
Meng, L., Srivastava, A., Kuča, K., & Wu, Q. (2022). Earthworm (Pheretima guillelmi)-mycorrhizal fungi (Funneliformis mosseae) association mediates rhizosphere responses in white clover. Applied Soil Ecology. https://doi.org/10.1016/j.apsoil.2021.104371
Zhang, W., Cao, J., Zhang, S., & Wang, C. (2016). Effect of earthworms and arbuscular mycorrhizal fungi on the microbial community and maize growth under salt stress. Applied Soil Ecology, 107, 214-223. https://doi.org/10.1016/J.APSOIL.2016.06.005.

Additionally, I recommend focusing the justification on the urban territories, which are seemingly understudied compared to natural grasslands.

LL. 113-120: I would suggest providing a hypothesis that has been tested instead of (or together with) giving a description of the experiment.

Experimental design

Despite the fact that the article uses a two-factor design (AMF*earthworms) often used in pot experiments, its distinctive feature is the evaluation of the effect on 5 turf grass species (3 gramineous and 2 leguminous species). The paper also presents and analyses data on soil properties in each of the experimental treatments.

Major comments on methods:

LL. 139-150 and throughout the MS: I strongly recommend avoiding using “standardized” abbreviations, like T1, T2, T3. If the factor should be named by abbreviation, please provide those that are more intuitive, like “E” or “EW” for worms, “AMF+EW” for combination, etc. This is especially poorly readable in the tables.

Validity of the findings

The Results seem valid, fitting the aims and scopes of the Journal. However, the representation of data, especially the PCA, should be improved. Please find the detailed comments in the attached file.

Major comments on Results:

Please consider changing the representation of PCA results from giving formulae to a clear output on the plot (figure) with centroids of your samples, vectors of your factors, etc. I would also recommend the Authors to perform, e.g., MANOVA on the individual PC pairs to find the optimal separation of centroids.

Major comments to the Discussion:

This section is generally well-written, taking into account the language quality that can be improved throughout the MS.
LL. 348-361: I’d suggest that authors deepen the discussion of this topic. What are the mechanisms behind observed effects? Which interactions happen between EW and AMF those and how? Given arguments with references are good, yet generally too scarce. Please check this review paper:
Pelosi, C., Taschen, E., Redecker, D., & Blouin, M. (2023). Earthworms as conveyors of mycorrhizal fungi in soils. Soil Biology and Biochemistry. https://doi.org/10.1016/j.soilbio.2023.109283.

Major comments to Conclusions:

I would recommend focusing conclusions on the problem of the study, and the hypotheses (which are not clearly given for now), approvals, or non-approvals.

Additional comments

Minor comments:

LL. 42-43: I’d recommend the Authors to briefly mention the “inconsistencies” by, e.g., naming those.
L. 47 and throughout the MS: Please verify that all Latin taxon names are in italics, e.g., Glomus mosseae
L. 49: “both the plant and the soil” -> both plants and soil
LL. 89-90: positive effect on what?
L. 124: Please never start the sentence with abbreviations. Please indicate that it’s an AM fungal strain
L. 126: Please briefly introduce the earthworm species by giving at least a couple of ecological characteristics
LL. 132-135: Is the chosen soil typical for urban territories? If now, how was the choice made? Please detail.
L. 155. Please describe the calculations before giving the formulae.
LL. 163-169: Please provide references for methods used.
LL. 170-186: In which environment was the PCA calculated? If it was done in SPSS, why are the formulae given? This method of representation is poorly understandable by the reader, as unnecessarily complicated.
LL. 188-191: Which variable was the response? Was the normality of data or models’ residuals approved before? How? Was ANOVA also performed in Excel or SPSS? Not clear.
L. 198: Please, here and throughout the MS (including table captions) - indicate the statistical output clearly, e.g. (ANOVA, Fx,y =…, p = 0.0XXX or p < 0.0001).

---

## Round 0.2 · Minor Revisions

· Academic Editor

Minor Revisions

Please have a look at the comments from the last reviewer. I think these are necessary to address.

Reviewer 1 ·

Basic reporting

This research contributes significantly by systematically evaluating how arbuscular mycorrhizal fungi (AMF) and earthworms (EW), individually and in combination, enhance soil health. The authors provide compelling evidence of their positive impact on five key plant-related soil functions and subsequent plant growth, making a strong case for the publication of this valuable research.

Experimental design

Experimental design has been updated by adding the AMF supplementation method and clarifying the control group.

Validity of the findings

The results have been further updated by adding a two-way ANOVA to find the interactive effects of AMF and earthworms.

Reviewer 2 ·

Basic reporting

I'm generally satisfied with the revisions made by the Authors, that basically resulted in re-writing of more than a half of the MS text, that, together with the spell-checking performed, significantly increased the quality of the paper. However, I've indicated some more minor moments to be revised before the MS can potentially be published. Please find those ion details in Section 4.

Experimental design

I'm generally satisfied with the revisions made by the Authors.

Validity of the findings

I'm generally satisfied with the revisions made by the Authors

Additional comments

Minor comments:

LL. 71-81: Maybe Urban lawns, not just lawns are to be introduced (spelling issue)

L. 87 and throughout the MS: please use correct spelling "arbuscular mycorrhizal fungi"

L. 89 and throughout the MS: Please rephrase to avoid beginning sentences with abbreviations. It's better to use full version of AMF in those cases.

L.90: relationships

L. 101: add space before opening bracket

L. 122 and throughout the MS: please carefully check the correctness of reference styles (e.g. not all letters should be capital)

L. 133: aimed at investigation...

LL. 133-153: some significant part of text is duplicated

Methods and throughout the MS: Please insure correct spacing of numbers from abbreviations, e.g. L. 179: 70.46 mg..

L. 213: please indicate whether H2O or saline pH was measured

Results section: please add F-values with DFs to all of the p-values provided throughout the text.

Section 2.3: I still suggest carefully moving the formulae to Supplementaries

Tables 1-3 can be represented as plots, and I would recommend turning those into simple figures (with mean and SDs differences in %% from the Control).

Acknowledgements section is now completely missing, while the initial version included information on funding. Please check and correct.

---

## Round 0.3 · accepted · Accept

· Academic Editor

Accept

Thank you for making the changes as requested by the reviewers.